# Cholesterol and Sphingolipid Enriched Lipid Rafts as Therapeutic Targets in Cancer

**DOI:** 10.3390/ijms22020726

**Published:** 2021-01-13

**Authors:** Michela Codini, Mercedes Garcia-Gil, Elisabetta Albi

**Affiliations:** 1Department of Pharmaceutical Sciences, University of Perugia, 06126 Perugia, Italy; michela.codini@unipg.it; 2Department of Biology, University of Pisa, 56127 Pisa, Italy; mercedes.garcia@unipi.it; 3Interdepartmental Research Center Nutrafood, Nutraceuticals and Food for Health, University of Pisa, 56127 Pisa, Italy

**Keywords:** cholesterol, sphingolipid, cancer, lipid raft, therapeutic target

## Abstract

Lipid rafts are critical cell membrane lipid platforms enriched in sphingolipid and cholesterol content involved in diverse cellular processes. They have been proposed to influence membrane properties and to accommodate receptors within themselves by facilitating their interaction with ligands. Over the past decade, technical advances have improved our understanding of lipid rafts as bioactive structures. In this review, we will cover the more recent findings about cholesterol, sphingolipids and lipid rafts located in cellular and nuclear membranes in cancer. Collectively, the data provide insights on the role of lipid rafts as biomolecular targets in cancer with good perspectives for the development of innovative therapeutic strategies.

## 1. Introduction

Over the last two decades, scientific research has conceptually changed the structure and function of the cell membranes described about 40 years ago. It was the faraway 1972 when Singer and Nicolson described the fluid mosaic theory, hypothesizing that the cell membrane is composed of a homogeneous lipid bilayer formed by glycerophospholipids and cholesterol, and inside proteins playing the main communication and response roles of the cell membrane. Over time, a mosaic of domains with unique biochemical compositions has been demonstrated, and among them, lipid rafts. Originally, lipid rafts were identified as ordered structures due to the aggregation of sphingolipids that were responsible for their molecular composition and properties, different from those of the surrounding membrane. Later, the presence of cholesterol within a liquid-ordered phase that interacted with sphingolipids by stabilizing lipid rafts was demonstrated.

## 2. Lipid Rafts

### 2.1. Lipid Rafts in Cell Membrane

In the cell membrane, lipid rafts or cholesterol-sphingolipids enriched domains act as platforms for cellular and/or exogenous proteins with different functions, including the shuttling of molecules on the cell surface, organization of cell signal transduction pathways, entry for pathogens and toxins, and the formation of pathological forms of proteins [1]. For example, sphingomyelinase influences cell proliferation and migration by regulating the ceramide/sphingomyelin balance in lipid rafts and thereby changing receptor-mediated signal transduction [2].

In the brain, lipid rafts are found in neurons, astrocytes, and microglia and play a role in the organization of multiprotein complexes involved in signal transduction. Cognitive decline in aging and neurodegenerative diseases is associated with the destructuration of lipid rafts [3]. Interestingly, gangliosides are enriched in lipid rafts and their levels change in healthy aging, Huntington’s disease, Alzheimer’s disease, Parkinson’s disease, amyotrophic lateral sclerosis, stroke, multiple sclerosis and epilepsy [4]. Among gangliosides, monosialoganglioside 1 (GM1) is the most common in lipid rafts of the brain. GM1 in lipid rafts mediate the pathophysiology of several cellular processes, including development, differentiation, and protection of neuronal tissue. GM1 has been proposed to play a negative role in Alzheimer’s disease by changing spatial organization of extracellular amyloid β [5].

In the inflammation process, lipid rafts contribute to permitting of the entry of bacterial pathogens into non-phagocytic cells, i.e., *Listeria monocytogenes*. In this way, the microorganism is capable of interacting and crossing physiological barriers, as intestinal, blood-brain, and placental barriers with severe tissue damage [6]. Moreover, some recognition receptors for viruses, including coronavirus, cluster into lipid rafts [7]. Thus, lipid platforms are considered as potential targets to block virus entry into endothelial cells [8].

### 2.2. Lipid Rafts in Nuclear Membrane

Over the past 15 years, significant research has focused on the presence cholesterol-sphingolipids enriched domains in the inner nuclear membrane. These lipid domains are now clearly appreciated to function as essential platforms for nuclear function. The inner nuclear membrane is a highly functional structure [9]. It is internally covered by the nuclear lamina associated with transcriptionally active chromatin, which includes the site for the nascent pre-mRNA transcripts. The “textbook” blueprint of nuclear structure and function describes the periphery of the nucleus and provides a platform for sequestering transcription factors away from chromatin, thereby modulating gene expression [10]. However, the reality is much more complex. The link of active chromatin (chromatin whose DNA is being duplicated or transcribed) to the nuclear lamina is due to specific lipid domains enriched in sphingomyelin and cholesterol [11]. In this way, sphingomyelin participates in maintaining the internal nuclear organization and function. Using sphingomyelinase, coupled with colloidal gold grains and immune anti-bromodeoxyuridine (BrdU), colocalization of sphingomyelin and nascent RNA has been demonstrated [12]. The intranuclear injection of sphingomyelinase destructures the active chromatin with alteration of the RNA transcription [12]. However, nuclear lipid domains contain transcription factors that regulate mRNA synthesis, i.e., signal transducer and activator of transcription 3 [11]. Interestingly, it has been demonstrated that both vitamin D3 and its receptor, which is a transcription factor, are localized in nuclear lipid domains [13,14]. Moreover, dexamethasone, a drug that acts directly in the nucleus by influencing RNA transcription, localizes in nuclear lipid domains [15].

## 3. Sphingolipids in Cancer

Sphingolipids have been identified as key regulators of a wide variety of cellular processes, including proliferation, differentiation, adhesion, autophagy, inflammation, migration and angiogenesis, which are crucial in cancer [16].

### 3.1. Sphingomyelin

Increased levels of sphingomyelin have been found in tumoral tissues such as hepatocellular carcinomas [17], compared to normal ones, but the specific changes depend on the type of cancers. However, sphingomyelin levels appear to be inversely correlated with the aggressiveness of some colorectal and breast cancer cell lines. Lower levels of sphingomyelin were found in prostate cancer, in nonsmallcell lung cancerand in esophageal tumor compared with their respective normal tissue due to the dysregulation of its biosynthesis [18]. Interestingly, sphingomyelin biosynthesis might be a downstream target of oncogenes and might also alter oncogenic activity. For example, upregulation of sphingomyelin synthase 1 transcription, translation and activity by the oncogene Bcr-abl has been reported in chronic myelogenous leukemia cell lines [19,20]. Pharmacological and siRNA-mediated inhibition of sphingomyelin synthase 1 in these cells reduced cell growth and induced the accumulation of ceramide, together with a reduction in diacylglycerol levels [19]. However, the maintenance of the correct localization and/or level of sphingomyelin at the plasma membrane is important to preserve the localization of oncogenic KRAS at the inner leaflet of the membrane and its protumorigenic effects [21]. Interestingly, the breakdown of sphingomyelin, due to the action of both acid and neutral sphingomyelinases, with the consequent production of different species of ceramides, is involved in the apoptosis of cancer cells. Notably, ingastric cancer cells gentamicin stimulates acid sphingomyelinase [22] and vitamin D3 neutral sphingomyelinase [23]. Moreover, vitamin D3 reduces glioblastoma cell aggressivenessvia neutral sphingomyelinase [24].

### 3.2. Ceramide

Ceramide is considered to be a mediator of cell death and senescence. Radiation treatment and chemotherapicdrugs, such as doxorubicin, etoposide, vincristine, and celecoxi band taxol, are known to induce ceramide accumulation, while decreased levels of ceramide are associated with drug resistance [25]. However, accumulating evidence indicates that distinct ceramide species, which are generated in different subcellular compartments and/or by different metabolic pathways, might exert different functions. Indeed, the effects of ceramides with specific chain lengths on cell fate appear to be cell-type specific. For example, a shift in ceramide composition from C24 to C16 increases the susceptibility to apoptosis in HeLa cells [26], but C16:0-ceramide generated by ceramide synthase 1 suppresses tumor growth, while C18:0-ceramide generated by ceramide synthases 5/6 protects tumors from apoptosis in human head and neck squamous cell carcinomas [27]. Celecoxib mediates its anti-proliferative effects, in part, by selectively activating ceramide synthase 6 in human colon carcinoma cells, leading to an increase in C16:0-ceramide [28]. Ceramide synthase 6 and C16-ceramide levels are higher in acute lymphoblastic leukemia cells compared to peripheral blood mononuclear cells and T lymphocytes derived from healthy human volunteers and it has been demonstrated that ceramide synthase 6 interferes with Fas-associated protein with death domain, death-inducing signaling complex (FADD DISC). FADD DISC assembly inhibiting the extrinsic pathway of apoptosis upon drug treatment, and thus inducing chemo-resistance in T-acute lymphoblastic leukemia cell lines [29]. The levels of C16:0-ceramide, C24:1-ceramide and C24:0-ceramide were found to be higher in malignant breast tumors than in benign and normal tissue and to correlate with disease severity [28]. By contrast, Moro et al. (2018) have found that the increased ceramide levels are inversely associated with aggressive phenotypes in breast cancers [30]. The authors also reported that a high ratio sphingosine-1-phosphate (S1P)/ceramide correlates with high proliferation and that patients with high expression of ceramide-generating enzymes exhibit worse prognosis, suggesting that this effect was probably due to the proliferative effect of S1P originating from ceramide. Acid ceramidase, the enzyme that catalyzes the formation of sphingosine, is over-expressed in 70% of head and neck squamous cell tumors compared with normal tissues, suggesting that this enzyme may play an important role in facilitating head and neck squamous cell cancer growth. Indeed, the over-expression of acid ceramidase in SCC-1 cells increases resistance to Fas-induced cell death, whereas down-regulation or chemical inhibition of acid ceramidase sesensitizes the SCC-1 cancer cell line to Fas-induced apoptosis [31]. Immunohistochemical analysis of primary prostate cancer samples showed that higher levels of acid ceramidase were associated with more advanced stages of this neoplasia [32]. Moreover, radiation-induced acid ceramidasee confers prostate cancer resistance and tumor relapse [33]. It has been reported that acid ceramidase promotes the nuclear export of phosphatase and tensin homolog (PTEN) through S1P- mediated Akt signaling, leading to tumor cell proliferation, and resistance to therapy [34]. Elevated expression of acid ceramidase in breast and epithelial ovarian cancers has been associated with positive outcomes [35,36].

### 3.3. Ceramide-1-Phosphate

Ceramide kinase (CerK) phosphorylates ceramide to form ceramide-1-phosphate (C1P), a molecule involved in proliferation, migration and inflammation [37] via the production of several cytokines and chemokines [38]. C1P-enhanced pancreatic tumor cell migration and invasion are dependent upon PI3K, Akt1, mTOR1, ERK1-2, and RhoA/ROCK signaling pathways [39]. CerK activity is required in the proliferation of neuroblastoma cells [40], of breast cancer cells [41], and of Kaposi sarcoma stem cells [42]. In metastatic breast cancer cells, CerK is upregulated and contributes to the migration and invasion through PI3K/Akt/Rho kinase activation [43]. By contrast, downregulation of CerK increases migration in A549 lung cancer cells by stimulating Rac1 activation and lamellipodium formation [44].

### 3.4. Sphingosine-1-Phosphate

Sphingosine 1-phosphate (S1P) is generated by sphingosine kinases 1 and 2 (SphK1 and SphK2) and is a bioactive lipid mediator, which acts both as a second and a first messenger. It is able to bind to five G-protein coupled receptors S1P1R-5 and to regulate different aspects of cancer progression, including cell proliferation, apoptosis, migration, inflammation and angiogenesis [16,45]. Many studies have suggested that the SphK1/S1P axis could play a role in the promotion of several types of tumors, based on the overexpression of SphK1 in tumor cell lines and in patient samples, and sometimes, on the measured increased S1P levels in tumors and/or in patient blood. These studies have recently been reviewed for breast [46], ovarian, [47], and gastrointestinal cancers [48], hepatocellular carcinoma [49], melanoma [50] and glioblastoma [51]. High S1P in breast tumor tissue is significantly associated with lymphnode metastasis [52]. In addition, S1P increased angiogenesis. Thus, in breast cancer, the SphK inhibitor decreases angiogenesis around the primary tumor and in lymph nodes [53]. From these observations, SphK/S1P molecules have been considered cancer biomarkers or therapeutic targets. However, the utility of targeting S1P in tumors is hindered by the role of S1P1 on immune cell trafficking [54]. Recently, it has been proposed that targeting S1PR4, which has a minor effect on immune cell trafficking, could be a successful alternative. Indeed, S1PR4 ablation reduces mammary and colorectal tumor growth and improves chemotherapy [55].

### 3.5. Gangliosides

Some *O*-acetylated gangliosides, which are transiently expressed during development, reemerge during tumorigenesis and can be used as tumor markers [56]. GM3 is expressed in melanoma, but not in normal melanocytes [57]. Expression of disialoganglioside 2 (GD2) and GD3 leads to cell growth and invasion [58]. GD2 might have a role in the fixation of melanoma cells at metastasized sites [59]. Moreover, it is highly expressed in neuroblastoma [60] and an anti-GD2 antibody Dinutuximab (Unituxin™) has been approved by Food Drug Administration and European Medicines Agency for the treatment of high-risk neuroblastoma patients [61]. GM3 is associated with aggressiveness and negative outcomes in many types of tumors, such as neuroblastoma, astrocytoma, thyroid carcinoma, sarcoma, cutaneous melanoma, and non-small cell lung cancer [62]. In addition, GM3 inhibits tumor cell proliferation through angiogenesis inhibition or decreases in cell motility [63].

## 4. Cholesterol in Cancer

Today, the dysregulation of cholesterol metabolism, pathologically associated with cancer, is the object of extensive discussion. Epidemiological studies reported an association between cancer and serum cholesterol levels but these results have not been confirmed by other authors [64]. Differences in the results might partly depend on the investigated tumour. A positive association between elevated serum cholesterollevels and increased cancer risk has been found in melanoma, prostate cancer, non-Hodgkin’s lymphoma, endometrial and breast cancer [65]. In other types of cancers, no association with hypercholesterolemia [66] or hypocholesterolemia [67] was found. Another crucial aspect about cholesterol in cancer relates to the diet. Even if several studies suggest a positive correlation between dietary cholesterol uptake and cancer risk, these studies are considered to be questionable, given the poor reliability of dietary investigations [68,69].

Interestingly, cancer development and progression is closely related to cholesterol metabolism at the cellular level. Less important is the blood cholesterol level [70,71]. Furthermore, studies on intracellular cholesterol traffic inhibitors showed the importance of subcellular cholesterol distribution [72]. Once synthesized in the endoplasmic reticulum, cholesterol is distributed to cell organelles, and to the plasma membrane, by sterol transfer protein [73]. Intracellular cholesterol levels and its compartmentalization are modulated by a complex and are integrated by a metabolic network involving de novo biosynthesis, the LDL-cholesterol entry by LDL receptors, cholesterol distribution by sterol transferring proteins, and cholesterol efflux by ABCA1 and ABCG1 proteins [74]. More than one gene and/or protein of this complex network can be deregulated in cancer. Indeed, it has been observed that many enzymes involved in the de novo cholesterol biosynthesis are deregulated in cancer cells, especially 3-hydroxy-3-methyl-glutaryl-coenzyme A reductase (HMGCR), the rate-limiting enzyme of the entire biosynthesis process [75,76]. Statins, a class of HMGCR inhibitors, have anti-cancer activity in a wide range of both liquid and solid tumours [77], not including invasive breast cancer in postmenopausal women [78]. Oncogenic and tumor suppressor factors activate or inhibit, respectively, sterol regulatory element-binding proteins, consequently regulating cholesterol synthesis [79]. Notably, the tumor suppressor protein p53 inhibits the sterol regulatory element-binding proteins-mevalonate pathway [80]. The analysis of more than 800 human cancer cell lines showed that inhibition of the mevalonate pathway by p53 activates mechanisms of tumor suppression in many types of cancer [81]. A role of the interaction between p53 and Hippo tumor-suppressor pathway in the coordination of cell proliferation has been reported [82]. In addition, several molecules derived from the cholesterol metabolism have also been implicated in the development of several types of cancer [83], including oxisterols [84], estrogens [85] and bile acids [86].

Moreover, a lot of experimental evidence underlines a close correlation between lipid rafts present in tumor-associated macrophages, i.e., macrophages that change phenotypes in the tumor microenvironment, and tumor progression and invasion [87]. This is relevant, considering that tumor-associated macrophages play a role in immunological suppression [88].

## 5. Lipid Rafts in Cancer

A large body of work has netted several significant advances in the study of the role of lipid rafts in cancer. These include molecular mechanisms that have led to the elucidation of key functions of sphingolipids and cholesterol in the cell membrane and nuclear membrane platforms. Unwittingly, these advances have highlighted previously unappreciated complexities of interactions between lipid and protein components of lipid rafts. Thus, an intensive 10 years of study has led to the discovery that well known proteins involved in migration, cell adhesion, invasion, angiogenesis and metastasis are localized in lipid rafts (Figure 1) [89].

A proteomic analysis of lipid rafts and cytoskeleton demonstrated a stronger protein–protein interaction network in breast cancer, melanoma, and renal carcinoma cells than in the respective normal cells [91]. Simulation analysis showed a higher stabilization of lipid rafts in cancer cells than in normal cells [91].

Significant research has focused on cholesterol and sphingolipids metabolism enzymes in lipid rafts and has demonstrated that they act as stimulus responders/coordinators, involved in the cell reaction to several pathophysiological conditions. Recently, cholesterol metabolism is clearly appreciated to function as a regulator of lipid rafts in cell membranes. Farnesyl-diphosphate farnesyltransferase 1, a key enzyme of cholesterol synthesis, increases lipid raft content in the membranes of cancer cells [92]. Goossens et al. (2019) demonstrated that, in ovarian cancer cells, cholesterol depletion from lipid rafts, caused by increased cholesterol efflux due to specific transporters, is responsible for the phenotypic reprogramming of macrophages into tumor-associated macrophages, making them more responsive to pro-tumor signals such as interleukin-4 and more resistant to the action of anti-tumor cytokines such as interferon-gamma [93].

Moreover, molecular localization of enzymes of sphingolipid metabolism in lipid rafts have led to elucidation of their key function in cancer cells and/or in drug sensitivity. SphKs and S1PRss are localized both inside and outside lipid microdomains [94]. SphK1 phosphorylation and translocation to plasma membrane microdomains is important for its oncogenic effects while localization outside the microdomains leads to survival with inhibition of proliferation [95]. GMs present in lipid rafts of cancer cells are involved in signaling, cell-to-cell recognition, and invasion [57,60]. Notably, GD3 accumulation in rafts has been associated with apoptosis induction [96]. GD3 is normally localized at the plasma membrane but, it translocates to the mitochondrial membranes following CD95/Fas-induced apoptosis [97]. It has been hypothesized that GD3, by interacting with mitochondrial raft-like microdomains, may trigger events associated with apoptosis, including changes in mitochondrial membrane potential, mitochondrial fission, and release of apoptogenic factors. Then, some GD3-containing “small” mitochondria, possibly derived from their fission process, could reach the nuclear envelope and contribute to the execution of apoptosis [98].

Thus, significant progress has been achieved in the past few years in understanding specific cholesterol-sphingolipid pathways mediated by specific enzymes. This progress is being translated in the development and optimization of novel therapeutic strategies based on targeting lipid rafts of cancer cell membranes [99].

### 5.1. Lipid Rafts in Metastasis

Many processes involved in cancer metastasis are modulated by lipid rafts. They include cell adhesion, migration and epithelial-to-mesenchimal transition (EMT), which is characterized by the loss of contact between cells and acquisition of fibroblastic phenotype. There are recent reviews on the role of lipid rafts in metastasis and EMT [89,90]. For example, disrupting lipid rafts with cyclodestrin or nystatin reverses EMT induced by TGF-β1 in breast cancer and gastric cancer cells, respectively [100,101]. Many studies demonstrate that proteins localized in lipid rafts such as podoplanin, caveolin-1, flotillins and CD44 are essential for EMT. For example, raft localization of podoplanin is necessary for the recruitment and activation of proteins of the Ezrin family, induction of EMT and thereby invadopodia function [102]. Upregulation of caveolin-1 is associated with EMT, increased migration of bladder cancer cell lines and metastatic bladder cancer [103]. Overexpression of caveolin-1 enhances EMT by increasing the transcription factor Slug, through activation of the PI3K/AKT pathway [103], while knockdown of caveolin-1 reduces Slug expression and EMT. Flotillins are upregulated in a variety of cancers [104], including gastric cancer. Flotillin-2 is is required for TGFβ-induced EMT in gastric cancer cells [105] since downregulation of flotillin-2 reduced the expression of the EMT markers vimentin and N-cadherin. CD44 is a biomarker of cancer stem cells, it is involved in the regulation of proliferation, invasion, metastasis and resistance and it is usually a marker of adverse prognosis [106]. The proteolytic cleavage of CD44 is required for the migration of tumor cells. CD44 localized in lipid rafts while its processing enzyme, the metalloproteinase 10, is mainly found outside rafts. Localization of CD44 outside lipid rafts in human glioblastoma allows metalloproteinase-mediated CD44 shedding and tumor cell migration [107] while raft disruption by cyclodestrins or simvastatin increase CD44 shedding and migration. Localization of CD44 in lipid rafts by palmitoylation limits the interaction of CD44 with its migratory binding partner ezrin, and reduces breast cancer cell migration [108].

Other proteins involved in invasion and metastasis, including potassium channels, urokinase-type plasminogen activator receptor and the type 1 transmembrane glycoprotein mucin 1, are found in lipid rafts [90]. The association of the potassium channel, SK3 with the calcium channel Orai1 in lipid rafts leads to the upregulation of calcium influx, increase of breast cancer cell migration and bone metastasis [109]. The treatment with the alkyl-lipid ohmline results in the translocation of the SK3–Orai1 complex out of lipid rafts, calcium entry reduction and cell migration impairment [109].

Interestingly, Tisza et al. [110] have reported that lipid raft destabilization is necessary for EMT. In breast cancer cells, this reduction in stability was necessary for the maintenance of the stem cell phenotype and EMT-induced remodeling. When raft stability was increased by adding docosahexaenoic acid, the metastatic capacity of the cells was reduced. Therefore, not only disrupting lipid rafts prevents EMT signaling, but EMT also requires modification of the physical properties of the lipid microdomains.

### 5.2. Lipid Rafts as Therapeutic Targets in Cancer

There are multiple lines of evidence that lipid rafts are promising targets in cancer therapy (Table 1).

One interesting feature to consider in anti-cancer therapy is the recent use of anti-cancer compounds with clinical efficacy as tumor necrosis factor-related apoptosis-inducing ligand (TRAIL) that selectively induces the apoptosis pathway in tumor cells leading to tumor cell death. However, some compounds have entered clinical trials in oncology [111]. However, they have had some problems in their application due to drug resistance, off-target toxicities, short half-life and limited uptake of TRAIL genes. Some interesting results have involved lipid rafts. Edelfosine, aalkylphospholipid analog, recruiting the death receptor into lipid rafts, is able to induce cancer cells apoptosis [112]. Moreover the anticancer effect of endocannabinoids is mediated by the perturbation of lipid rafts [113]. Thus, the recruitment of death receptors into lipid rafts is essential for drug response through apoptosis activation in cancer cells. However, the involvement of lipid raft is not limited to death receptors, but includes other signaling molecules [114]. Upregulation of lipid and protein metabolism enzymes of lipid rafts is involved in drug transport [115]. Gajate and Mollinedo (2015) reported that the presence of Fas/CD95 in lipid rafts is a putative target for hematologic cancer therapy [116]. These data suggest that the quantity and structure of cellular and nuclear lipid rafts are important for the therapeutic response of cancer cells. This emerging role of lipid rafts has significant implications in different liquid and solid tumors.

Lipid rafts are involved in liquid and solid cancers (Figure 2).

#### 5.2.1. Liquid Tumors

Lipid rafts were shown to control the drug response of patients affected by B-chronic lymphocytic leukemia. It has been demonstrated that the susceptibility of patients with B-chronic lymphocytic leukemia is linked to the embedding of the promoting death receptor in lipid rafts and its exclusion is responsible for TRAIL resistance [117]. In non-Hodgkin’s T cell human lymphoblastic lymphoma cells, serum cholesterol reaches inner nuclear membrane by enriching it with lipid rafts content and regulating gene expression with the overexpression of antioxidant proteins as superoxide dismutase 1 and 2, copper chaperone for superoxide dismutase, peroxiredoxins 1, glutathione reductase, glutathione synthase, catalase and polynucleotide kinase/phosphatase [132]. In the same cells, gentamicin inhibits nuclear nSMase and stimulates SM-synthase by increasing SM content and thereby making nuclear lipid rafts more rigid structures useful in regulating gene expression [130].

#### 5.2.2. Solid Tumors

Several recent studies have disclosed specific functional and patho-biological roles of lipid rafts in solid tumors, including breast, stomach, liver, and pancreas cancer [118,119,120,121,122,123,124,125,126,127,131,133,134,135,136,137,138,139,140,141,142,143].

#### 5.2.3. Breast Cancer

The multiple advances in breast cancer have led to a better understanding of the molecular mechanisms that occur in lipid rafts involved in migration, invasion, angiogenesis, and metastasis. A proteomic study on lipid rafts purified from breast cancer cells showed an abundance of groups of proteins specific for the progression toward malignancy [133]. Promotion of breast cancer cell migration by TNF-α-MAPK/ERK signaling pathway requires translocation of membrane-type metalloproteinase 1 and matrix metalloproteinase 2 proteins into lipid rafts [134]. Moreover, localization of the epidermal growth factor receptor in lipid rafts induces activation of Akt epidermal growth factor receptor kinase-independent [135]. However, downregulation of caveolin 1 in lipid rafts of lysosome is responsible for enhanced autophagy in human breast cancer cells and tissues [136].

As a consequence of the interest in lipid rafts of breast cancer cells, the study of pharmacological targets has also expanded. The liver X receptor that is implicated in the cholesterol metabolism modulates lipid rafts integrity with consequent inhibition of breast cancer cell growth [118]. It has been demonstrated that anticancer activity of γ-tocotrienol is not only due to its accumulation in the lipid rafts, resulting in disrupting of HER2 dimerization and activation in lipid rafts [119], but also to the reduction of heregulin content in exosome with subsequent decrease in autocrine/paracrine mitogenic stimulation [137]. Interestingly, bufalin stimulates the entry of death receptor 4 (DR4) and DR5 into lipid rafts by promoting breast cancer cell apoptosis. In this way, the cells that were TRAIL resistant became TRAIL sensitive [120]. Li et al. (2017) showed that miR-3908 is involved in the migration process via lipid rafts that act as platform for interaction between adiponectin receptor 1 and Flotillin-1 [138]. Thus, the authors suggest that targeting miR-3908 and the lipid raft, might be the correct way to prevent and/or treat breast cancer. Later is has been shown that resveratrol changes the distribution of flotillin-1 and flotillin-2 in lipid rafts with consequent anticancer effects [121]. The use of methyl beta cyclodextrin, which extracts cholesterol from lipid rafts destroying them, has permitted the identification of the location in lipid rafts of urokinase-type plasminogen activator receptor and matrix metalloprotease-9 essential for breast cancer cell migration, invasion and angiogenesis [139]. Nano assembly of methyl beta cyclodextrin with hyaluronic acid-ceramide, which targets the CD44 receptor and destroys lipid rafts, has been developed for breast cancer therapy [122].

#### 5.2.4. Gastric Cancer

Gastric cancer cells are insensitive to TRAIL, but inhibition of epidermal growth factor receptor activation and redistribution in lipid rafts can increase the sensitivity and therefore stimulate apoptosis [123]. The anti-epidermal *growth factor receptor* monoclonal antibody cetuximab facilitates the action of TRAIL in gastric cancer cells by promoting death receptor 4 clustering and FADD protein translocation into lipid rafts [123]. A drug extracted from traditional Chinese medicinal herb, β-elemene, by acting on lipid rafts, reduces the drug resistance to TRAIL by inhibiting proliferation and stimulating apoptosis in gastric cancer cells [124].

#### 5.2.5. Liver Cancer

In hepatocarcinoma cells, localization of CD44 in lipid rafts is promoted by cholesterol with consequent inhibition of invasion and metastasis [140]. Sphingomyelin of lipid rafts localized in inner nuclear membrane of hepatoma cells is richer in palmitic acid than that of hepatocytes. It results in different dynamic properties of nuclear lipid rafts in the two cell types and increased import-export of protein signaling molecules in cancer cells respect to normal cells [141]. In the same nuclear rafts, daunorubicin changes the neutral sphingomyelinase activity, resulting in the saturated long chain fatty acid sphingomyelin decrease. Thus, nuclear lipid rafts change their function as platforms for active chromatin and for transcription process [131]. Fumonisin B_1_, a mycotoxin that inhibits Cer-synthases, promotes liver cancer by reducing sphingomyelin and by increasing free cholesterol with the consequent modulation of lipid rafts [125]. Inhibition of Toll-like receptor 7, a molecule belonging to a family of receptors playing an important role in innate immune responses, with specific antagonist 20S-protopanaxadiol changes caveolin-1 and flotillin-1 molecules in lipid rafts of hepatic cancer cells delaying, consequently, cell proliferation and migration [126].

#### 5.2.6. Pancreatic Cancer

CD133 pancreatic cancer marker co-localizes with signaling proteins for cancer aggression in lipid rafts [142]. Lovastatin by inhibiting mevalonic-acid-pathway in cholesterol synthesis reduces chemoresistance in CD133^Hi^ cells and in CD133^lo^ cells [127]. Recently, it has been reported that localization of complement component 1, q subcomponent binding protein (C1QBP) in lipid rafts regulates the liver metastatic process induced by IGF-1/IGF-1R [143]. The authors suggest that targeting C1QBP in lipid rafts could be a strategy to limit hepatic metastasis [143].

#### 5.2.7. Others

The attention of lipid rafts in lung cancer has been focused in non-small cell lung cancer cell lines. The cells resistant to gefitinib have higher cholesterol levels than the gefitinib-sensitive cell lines [144]. Squalene synthase, a key enzyme for the cholesterol synthesis, enriches the content of tumor necrosis factor receptor 1 in lipid rafts, thereby cancer metastasis. Activation of c-Met in rafts represents a key metabolic pathway responsible for resistance to radiation [145]. Additionally, in endometrial cancer cells, CD24-mediated amplification of the Met signaling is responsible for drug resistance [146].

The higher cholesterol content in prostate cancer cells than in results in increased lipid raft formation and regulation of tumor progression [147]. In lipid rafts of prostate cancer cells CXCL12 interacts with its receptor CXCR4, a G protein coupled receptor. CXCL12/CXCR4 interaction transactivates EGRF2/HER2 stimulating invasion [128].

Pirmoradi et al. (2019) have suggested that targeting cholesterol metabolism in glioblastoma cells might be promising since CHO regulates temozolomide-induced cell death due to the accumulation and activation of death receptor 5 in lipid rafts [129].

In renal cell cancers, GalNAc-disialyl Lc4 forms a molecular complex with integrin β1 and caveolin-1 in lipid rafts and increases malignant properties that can be reverted by antibodies [148].

## 6. Conclusions

In this review, we have attempted to highlight novel insights into the role of lipid rafts in cancer cells. The involvement of lipid rafts in signaling processes modulating proliferation, death and metastasis suggests that they might be promising targets in cancer therapy. Changes of rafts by manipulation of sphingolipid and cholesterol content or by antibodies against gangliosides alter the signaling of the growth factor and death receptors localized inside. Chemotherapy drugs may act on death receptor clustering and/or disrupting receptor signaling. In the last years, the existence of lipid rafts in different cell compartments such as plasma membrane, mitochondrion and nucleus and their involvement in apoptosis and in the modulation of gene expression has become clear in normal and cancer cells. However, further investigation is necessary to understand the possible communication between rafts of different subcellular organelles and how this can be used to improve cancer therapy.

## Figures and Tables

**Figure 1 ijms-22-00726-f001:**
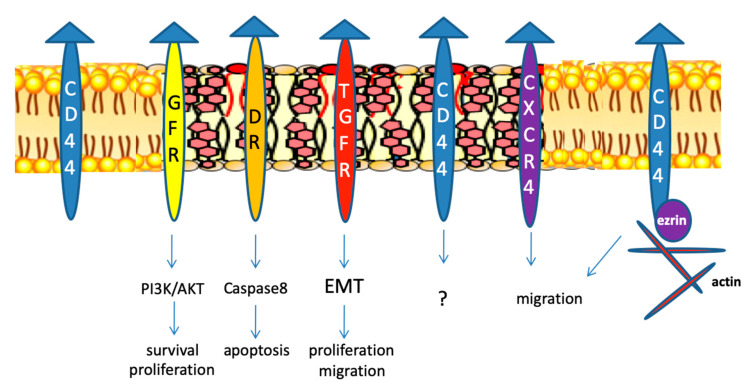
Roles of lipid rafts in cancer NO. Lipid rafts are involved in proliferation, apoptosis, migration, cell adhesion, and metastasis. Many proteins involved in cancer are localized in rafts. For example, growth factors receptors (GFR) through activation of PI3K/AKT or MAPK pathways lead to survival and proliferation; dead receptors (DR) such as TRAIL and Fas activate caspase 8 and the apoptotic cascade; the chemokine receptor CXCR4 activation leads to migration; the transforming growth factor-1 β receptor (TGFR) is involved in epithelial to mesenchimal transition (EMT). CD44 is localized in rafts, but its role is unclear (2). When it moves out of them, it is able to interact with the metalloproteinase 10 [90] and with ezrin, leading to actin remodelling and cell migration. Triangles: ligands of the receptors. See explanations and references in the text.

**Figure 2 ijms-22-00726-f002:**
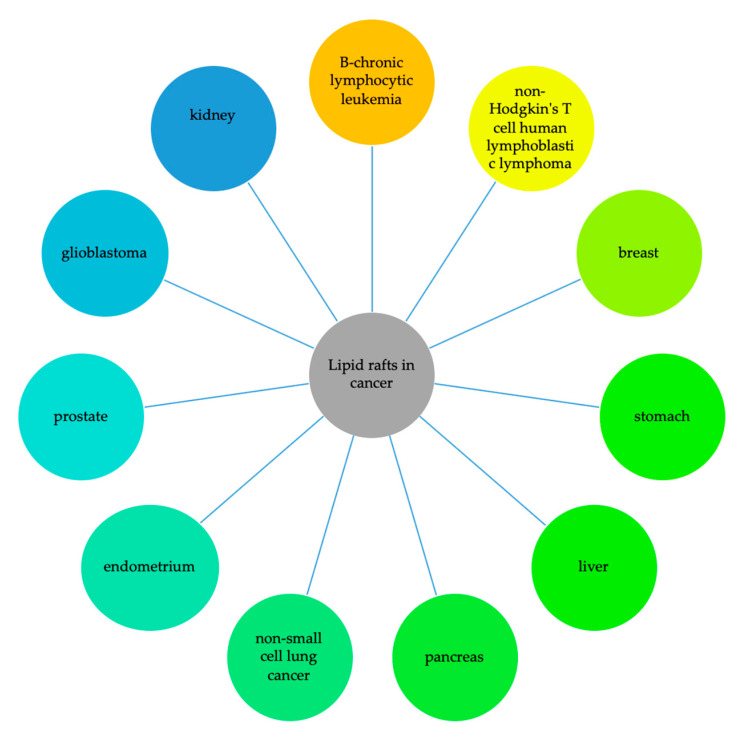
Schematic representation of the main cancers in which lipid rafts are involved. Lipid rafts are involved in liquid [117,130,132] and solid tumors [118,119,120,121,122,123,124,125,126,127,128,129,131,133,134,135,136,137,138,139,140,141,142,143,144,145,146,147,148] including tumors of breast [118,119,120,121,122,133,134,135,136,137,138,139], stomach [123,124], liver [125,126,131,140,141], pancreas [127,142,143], lung [144,145], endometrium [146], prostate [128,147], nervous system [129] and kidney [148].

**Table 1 ijms-22-00726-t001:** Main papers showing lipid rafts of cellular and nuclear membranes as targets of anticancer molecules.

MOLECULES TARGETING LIPID RAFTS IN CANCER	
Cell Membrane Lipid Rats	
Hydroxypropyl-β-cyclodextrin	[100]
Nystatin	[101]
TGF-β1	[100,101]
Podoplanin	[102]
Caveolin-1	[103]
Flotillin	[104,105]
CD44	[106,107,108]
SK3	[109]
Orai1	[109]
TRAIL	[111,117]
Edelfosine	[112]
Endocannabinoids	[113]
Liver X receptor	[118]
γ-tocotrienol	[119]
Bufalin	[120]
Resveratrol	[121]
Methyl beta cyclodextrin	[122]
Epidermal growth factor receptor	[123]
β-elemene	[124]
Fumonisin B1	[125]
Antagonist 20S-protopanaxadiol	[126]
Lovastatin	[127]
CXCL12	[128]
Temozolomide	[129]
Nuclear membrane lipid rafts	
Gentamicin	[130]
Daunorubicin	[131]

## Data Availability

Not applicable.

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
