# Peer review of "Cholesterol and Sphingolipid Enriched Lipid Rafts as Therapeutic Targets in Cancer"

_ijms, 2021, doi:10.3390/ijms22020726_

Round 1
Reviewer 1 Report
This is a well written review article detailing lipid rafts therapeutic targets in cancer.
- Spell check in the title (rafts)
- authors have detailed the role of lipid rafts in cancer and how they regulated in cancers. If possible can you please include a paragraph on one of the important steps in cancer progression i.e. Metastasis. Does lipid rafts have any role in this process? also include role lipid rafts role in Epithelial mesenchymal transition. Both these process requires changes in cell membrane.
Author Response
We thank the referee for this suggestion that improves the manuscript and apologize for the mistake in the title.
- The title has been corrected
- A paragraph on rafts and metastasis, including the role of rafts in epithelial to mesenchimal transition has been added (from p.6 line20 -to p.7 line 13). New references 99-110 have been included.
Reviewer 2 Report
The paper by Codini et al. is useful. However, there are several points, which should be changed. 1. The authors claim that the data presented in their review provide insights on the role for the role of lipid rafts as biomolecular targets in cancer with good perspectives for development of innovative therapeutic strategies. However, I could understand the role of lipid rafts in cancer. For instance, describing lipid rafts in the nuclear envelope the authors write that the link of active chromatin to the nuclear lamina is due to specific lipid domains enriched in SM and CHO. The intranuclear injection of SMase destructures the active chromatin. The nuclear lipid rafts contain signal transducer and activator of transcription 3 and regulate mRNA synthesis. They mentioned platforms for vitamin D and drug receptors. All these sentences are unclear. What does it mean the active chromatin?
They write that intracellular CHO levels are more implicated than blood CHO levels in development and progression of cancer suggesting that tumorigenesis is triggered by alteration in the cellular CHO homeostasis. What does it mean? It seems that the authors used "copy-past" without understanding or at least clarification. All these things should be explained. I could no find novel insights into the role of lipid rafts in cancer cells. There are only parts of abstracts of the papers.
- Why the authors extensively use abbreviations (even for one word: CHO; SMase...)? These abbreviations do not decrease the number of words in the text. Unnecessary abbreviations should be eliminated.
- The scheme describing the effect of lipid rafts on cancer should be included.
I suggest a major revision.
Author Response
- The section 2.2 has been rewritten. We hope that now the meaning of active chromatin (chromatin whose DNA is being duplicated or transcribed) is clearer for the reader (p.2 lines 13-34).
- The paragraph on cholesterol of section 4 has been simplified (p.5 lines 3,4).
3) We have eliminated the abbreviations for ceramide, cholesterol, sphingolipids, sphingomyelin and sphingomyelinase.
4) A new figure describing the effect of lipid rafts on cancer has been included .
Round 2
Reviewer 2 Report
I have no objections any more.